# Intertransverse process block to improve quality of recovery and pain management in cardiac surgery: Protocol for a double-blinded randomized controlled trial

Henry Man Kin Wong[1]*, Ranjith Kumar Sivakumar[1], Wai Tat Wong[1], Albert Kam Ming Chan[2], Zion Ho Sum Yeung[2], Pik Yu Chen[2], Sherry Tsz Wai Tang[2], Mandy Hiu Man Chu[1], Randolph Hung Leung Wong[3], Kwok Ming Ho[1]

1 Department of Anaesthesia and Intensive Care, The Chinese University of Hong Kong, Hong Kong SAR, China, 2 Department of Anaesthesia, Pain and Perioperative Medicine, Prince of Wales Hospital, New Territories, Hong Kong SAR, China, 3 Department of Surgery, Division of Cardiothoracic Surgery, Prince of Wales Hospital, New Territories, Hong Kong SAR, China

* henrymkwong@cuhk.edu.hk

## Abstract

### Background

Chronic postsurgical pain (CPSP) after cardiac surgery is significant. Despite the known association between acute pain and CPSP, advanced pain management strategies have not reduced its incidence. Preventing CPSP requires optimizing acute pain control and disrupting central sensitization. The side effects and risks associated with chronic use of current opioid-based cardiac anesthesia necessitate the adoption of multimodal analgesia. Regional anesthesia is a promising alternative, though existing techniques in cardiac surgery have notable limitations. The intertransverse process block (ITPB) is a novel regional technique that offers potential somatic and visceral analgesia. Recent studies demonstrate consistent local anesthetic spread to the intercostal, paravertebral, and epidural spaces, suggesting broader pain control. ITPB may provide a simpler, safer approach in cardiac surgery, reducing the risks of pleural puncture and bleeding. We hypothesize that ITPB will improve quality of recovery, pain control, and health-related quality of life, thereby mitigating chronic postsurgical pain.

### Methods

This is a single-center, randomized, double-blinded, placebo-controlled trial with intention-to-treat analysis. Elective patients awaiting coronary artery bypass grafting, with or without valvular repair or replacement, will be recruited. Ninety-six participants will be randomly assigned to either ITPB or control group. The ITPB group will receive bilateral ITPBs with 20 ml 0.25% levobupivacaine on each side at the T4-5

**Data availability statement:** No datasets were generated or analysed during the current study. All relevant data from this study will be made available upon study completion.

**Funding:** The author(s) received no specific funding for this work.

**Competing interests:** The authors have declared that no competing interests exist.

level under ultrasound guidance, administered before anesthesia induction. Sham blocks, with equal volume of normal saline, will be performed in the control group. The primary outcome is the quality of recovery, assessed using the 15-item Quality of Recovery questionnaire, at 24 hours after tracheal extubation. Secondary outcomes include Numerical Rating Scale pain scores, patient satisfaction, and opioid consumption within 72 hours post-extubation, duration of mechanical ventilation, length of stay in the ICU and hospital, and opioid-related side effects. The incidence of CPSP at 3, 6, and 12 months will be measured, along with pain interference via the Brief Pain Inventory and the Short-Form McGill Questionnaire-2.

## Discussion

Current pain management strategies often rely heavily on opioids, which can have significant side effects and may not adequately address chronic postsurgical pain. This study investigates the efficacy of the intertransverse process block, a novel regional anesthesia technique, in reducing both acute and chronic postsurgical pain in cardiac surgery. Randomized controlled trials on intertransverse process block in cardiac surgery are limited. The results of this study will help define the role of intertransverse process block on the recovery process, and generate vital preliminary data on its potential long-term benefits in reducing chronic postsurgical pain in cardiac surgical population.

## Clinical trial registration

This trial has been prospectively registered at clinicaltrials.gov: NCT06946290

---

## Introduction

The cornerstone of modern pain management in cardiac surgery lies in proactively preventing the development of chronic postsurgical pain (CPSP). This paradigm shift emphasizes pre-emptive interventions to disrupt the cascade of events and central sensitization that transform acute postoperative pain into a persistent, debilitating condition [1–4]. CPSP, a significant and often underestimated consequence of cardiac surgery, affects a substantial proportion of patients, with incidence ranging from 30–50% at 3 months, and up to one year after surgery [5–8]. Its impact extends beyond discomfort, significantly impairing daily function, reducing quality of life [8,9], and imposing a substantial economic burden on the healthcare system.

The pathophysiology of CPSP involves a complex interplay of factors, with central sensitization playing a pivotal role [3]. This process, characterized by heightened neuronal excitability within the central nervous system, amplifies pain signals and can lead to persistent pain even after the initial injury has healed. Therefore, a crucial aspect of CPSP prevention lies in disrupting the afferent nociceptive signals transmitted from injured tissues to the spinal cord and brain, thereby preventing the establishment and perpetuation of central sensitization [1,4].

While opioids have traditionally been the mainstay of postoperative pain management, their use is not without significant drawbacks. Opioids can have dose-dependent side effects, including respiratory depression, nausea, and constipation. Furthermore, prolonged opioid use can lead to tolerance, opioid-induced hyperalgesia, and an increased risk of chronic opioid dependence [10–12]. Recognizing these limitations, current pain management strategies emphasize multimodal analgesia, incorporating non-opioid medications and regional anesthesia techniques to optimize pain control while minimizing opioid-related risks.

Non-steroidal anti-inflammatory drugs (NSAIDs) offer a valuable adjunct in pain management, but their use in cardiac surgery is often limited due to concerns about bleeding complications and potential renal impairment. Other non-opioid analgesics, such as paracetamol and gabapentinoids, have shown limited efficacy in managing the intense pain associated with sternotomy [13]. Regional anesthesia techniques have emerged as promising strategies for both acute pain management and potential CPSP prevention in various surgical settings [14–17]. Techniques such as epidural anesthesia and paravertebral blocks have demonstrated efficacy in reducing postoperative pain intensity and opioid requirements [14,15]. However, their application in cardiac surgery presents unique challenges. Epidural anesthesia carries the risk of neuraxial haematoma due to systemic anticoagulation and heparinization, while paravertebral blocks may be associated with complications such as pneumothorax and pleural puncture [18]. Erector spinae plane block (ESPB) has shown inconsistent results in reducing postoperative pain and morphine consumption in cardiac surgery [19–24]. While one recent study found that is reduced pain and opioid consumption, it did not significantly affect the incidence or severity of CPSP [25]. Parasternal plane blocks offer advantages over neuraxial techniques [26–27] but may not adequately address visceral pain.

While regional anesthesia is crucial for managing acute postoperative pain, its impact on CPSP remains largely unknown. The potential for regional anesthesia to reduce CPSP has been identified as one of the top research priorities in anesthesia and perioperative care [28]. Intertransverse process block (ITPB) is a novel and promising alternative, targeting the paravertebral space through extra-paravertebral injection within the intertransverse tissue complex, posterior to the superior costotransverse ligament (SCTL) [29,30]. Recent MRI studies have demonstrated consistent spread of local anesthetic to the ipsilateral intercostal, paravertebral spaces, neural foramina, and epidural space following ITPB [31], suggesting potential for both somatic and visceral analgesia. ITPB has shown preferential spread of local anesthetic to the epidural space and neural foramina compared to ESPB, and effective analgesia in breast and video-assisted thoracoscopic surgeries [32,33]. Compared to other regional techniques in cardiac surgery, ITPB may offer a simpler and safer approach with reduced risk of pleural puncture and bleeding. This trial will assess whether ITPB improves quality of recovery after surgery and mitigates acute and chronic postsurgical pain in cardiac surgical patients. We hypothesize that ITPB will improve postoperative pain control and recovery, potentially reducing CPSP and improving long-term health-related quality of life.

## Materials and methods

### Study population and design

The protocol of this study follows the SPIRIT checklist (Supporting information S1 File). This is a single-centre, double-blinded, randomized controlled trial. Ethical approval was obtained from the Joint Chinese University of Hong Kong-New Territories East Cluster Research Ethics Committee on 22nd May 2023 (CREC Ref No 2025.177-T). The study is registered at http://clinicaltrials.gov (NCT06946290), with the registration date of 23rd April 2025. With an annual volume of approximately 400 elective cardiac cases, the recruitment is anticipated to start on 1st November 2025 and to be completed by 31st October 2026. Data collection will be completed by 31st October 2027, and the results will be expected by 30th December 2027. Eligible patients will be those undergoing elective cardiac surgery at Prince of Wales Hospital, a university teaching hospital with 1,650 beds in Hong Kong. Following surgery, all patients will receive initial postoperative care in a 28-bed ICU, where they will have continuous 1:1 nursing care. The anticipated discharge to a high-dependency cardiac ward is within 24 hours after surgery.

## Randomization and concealment

Randomization will be performed in 1:1 ratio using the REDCap randomization module in the study database, allocating participants to either the ITPB (intervention) or control group in randomly permuted blocks of size four. Sequentially numbered, coded, sealed, opaque envelopes, each containing the group assignment of either interventional or control are then prepared by a third party who takes no further part in the study. These envelopes are kept in a secure location and are only opened when a participant is enrolled and ready for the procedure. The ITPB syringes will be prepared by a nurse, who is not part of the patient care team, and not involved in the study, under sterile conditions. These syringes, which will contain either local anesthetic or saline, will be identical in appearance and labeled only with a blind code corresponding to the randomization list. Standardized surgical procedures will be executed by a surgical team that is blinded to the group allocation. Concurrently, personnel responsible for data collection in the intensive care unit and hospital wards, including anesthesiologists and nurses, will also be blinded to the treatment assignments.

## Eligibility criteria

Participants will be adults aged 18 and above who are undergoing CABG, valve repair/replacement, or combined CABG/valve surgeries via sternotomy. Patients will not be eligible if they are undergoing emergency, aortic, or redo surgery, have pre-existing chronic pain or are using chronic opioids/sedatives. Additional exclusion criteria include preoperative renal failure (either requiring renal replacement therapy or having a creatinine clearance $<30\,ml.min^{-1}$ by Cockcroft-Gault formula), liver enzymes levels twice the upper normal limit, a left ventricular ejection fraction less than 40%, a need for mechanical circulatory support, intraoperative remifentanil administration, or an inability to provide informed consent.

## Anesthesia and interventions

Following standard cardiac surgery monitoring, general anaesthesia will be initiated using midazolam ($0.01–0.05\,mg.kg^{-1}$), fentanyl ($2–5\,\mu g.kg^{-1}$), and rocuronium ($0.5–1\,mg.kg^{-1}$) to facilitate intubation. Anaesthesia will then be continued with a combination of sevoflurane and propofol infusion, with the goal of maintaining a Bispectral Index between 40 and 60. ITPB will be performed after anesthesia induction, with the patient in the lateral decubitus position. Intraoperative opioids (fentanyl and morphine) will be administered at the discretion of the anesthesiologist. Postoperative analgesia will be identical in both groups, including patient-controlled analgesia (PCA) morphine (one milligram bolus, lockout period of five minutes, 4-hour maximum dose of 25 mg) for 72 hours after surgery, oral analgesics (paracetamol 1 g every six hours, dihydrocodeine 30 mg thrice a day), and on-demand antiemetics (intravenous ondansetron 4 mg every eight hours). Rescue analgesics may be prescribed as needed. Upon arrival in the ICU, the propofol infusion will be discontinued to aid in weaning patients from the ventilator using Adapative Support Ventilation (ASV), a system that adjusts settings according to each patient's lung mechanics and breathing effort. Pain levels will be routinely evaluated in both the ICU and the ward. After extubation, specific pain assessments will occur at 2, 4, 8, 12, 24, 48 and 72 hours. Patients experiencing moderate to severe pain will be prescribed PCA morphine. Any instances of nausea, vomiting, and the administration of rescue antiemetics will be recorded.

## Ultrasound-guided block technique

The intervention group will receive bilateral ITPB prior to anesthesia induction, while the control group will receive sham blocks. All blocks will be performed by an experienced anesthesiologist who has completed more than fifty successful ITPB procedures and takes no further part in the study, using a Philips EPIQ ultrasound system with a curved array transducer (C5-1) and 80 mm echogenic nerve block needle (SonoTAP; PAJUNK, Germany). ITPB will be performed with patients positioned in the lateral decubitus position. The target intervertebral level (T4-5) will be identified and marked using a preview ultrasound scan. The transducer will be placed two to three centimetres lateral to the spinous process. Under strict asepsis, a

single-level (T4-5) ultrasound-guided ITPB will be performed using in-plane needle insertion from lateral to medial, targeting the medial aspect of the retro-SCTL space. Correct needle position will be confirmed by distension of the retro-SCTL space following a test bolus of 1–2 ml of 0.9% normal saline. Then, 20 ml of 0.25% levobupivacaine or 0.9% normal saline will be injected in small aliquots. The same procedure will be repeated on the contralateral side using the same volume of study medication. The time required to perform the block, from needle insertion to removal, will be recorded.

## Outcome measures

**Primary outcome.** Although acute pain is recognized as an important predictor for the development of CPSP, different acute pain management strategies have failed to reduce the incidence of CPSP, suggesting a complex mechanistic link between acute and chronic pain after surgery. Evidence indicates that pain-related functional interference and patient-reported outcomes, such as quality of recovery, may be associated with the development of CPSP [34]. Therefore, the primary outcome of this study is the quality of recovery, measured by the 15-item Quality of Recovery questionnaire score (QoR-15) at 24 hours after tracheal extubation. QoR is recommended for assessing patient comfort after surgery and is a highly valid and reliable patient-centered outcome measure [35].

**Secondary outcomes.** Secondary outcomes include Numerical Rating Scale (NRS) pain scores at 2, 4, 8, 12, 24, 48 and 72 hours after tracheal extubation, patient satisfaction with pain management, postoperative morphine consumption at the above time points, time to first morphine rescue (in minutes), intraoperative opioid consumption (converted into morphine equivalent doses), duration of mechanical ventilation, length of stay in ICU and hospital, opioid-related side effects, such as postoperative nausea and vomiting (PONV), incidence of CPSP at 3, 6, and 12 months, and pain interference assessed using the Short-Form McGill Pain Questionnaire-2 (SF-MPQ-2) and the Brief Pain Inventory (BPI) Interference Scale at 3, 6, and 12 months postoperatively.

The International Association for the Study of Pain (IASP) and International Classification of Diseases (ICD-11) defines CPSP as pain that develops or increases in intensity for at least 3 months after a surgical procedure [36]. Participants reporting CPSP will be further assessed for severity and impact based on the recommendations from the Initiative on Methods, Measurement, and Pain Assessment in Clinical Trials (IMMPACT). This includes the SF-MPQ-2 to assess the sensory pain qualities and affective components, and the BPI to assess pain interference with physical functioning.

The Chinese version SF-MPQ-2 [37] will be used to measure sensory and affective aspects of pain. It evaluates chronic pain symptoms on an 11-point numerical rating scale (0 = none, 10 = worst possible). It includes three sensory descriptors and one affective descriptor. The four subscales will be calculated as a mean of items in each subscale, and the total score will be the mean of all 22 items. Higher scores indicate more intense symptoms. The Chinese version of the Brief Pain Inventory (BPI) Interference Scale [38] will be used to evaluate the extent to which pain interferes with various aspects of functioning, including physical and emotional functioning, and sleep.

## Data collection

After eligibility screening, patients will receive an information sheet explaining the trial. A research nurse will discuss the study with them before obtaining written informed consent. Research team members, blinded to group allocation, will collect all data, including patient demographics and body mass index. Cumulative opioid consumption and time to first morphine rescue will be extracted from the PCA pump. Pain scores (NRS 0–10, where zero is no pain and 10 is worst imaginable) at rests and during coughing will be assessed at 2, 4, 6, 8, 12, 24, 48 and 72 hours post-extubation. Patients will use a verbal analogue scale (0–100, 0 = worst possible, 100 = best possible) at specified intervals to rate their overall satisfaction with pain management. Nausea, vomiting and any administration of rescue antiemetic use will be documented. The validated Chinese version of the QoR-15 questionnaire [39] will be administered at baseline (preoperatively) and at 24 and 72 hours postoperatively after tracheal extubation. The SF-MPQ-2 and BPI will be used to assess CPSP at 3, 6, and 12 months postoperatively (Fig 1).

| TIMEPOINTS | STUDY PERIOD | | | | | | | | | | |
| | Enrolment | Post-allocation | | | | | | | | | |
| | Baseline | 2h extubate | 4h extubate | 6h extubate | 8h extubate | 12h extubate | 24h extubate | 48h extubate | 72h extubate | 3-month | 6-month |
| **ENROLMENT:** | | | | | | | | | | | |
| **Eligibility screen** | X | | | | | | | | | | |
| **Informed consent** | X | | | | | | | | | | |
| **Demographic data** | X | | | | | | | | | | |
| **Comorbidity data** | X | | | | | | | | | | |
| **EuroScore** | X | | | | | | | | | | |
| **Allocation** | X | | | | | | | | | | |
| **INTERVENTION** | | | | | | | | | | | |
| **ITPB** | X | | | | | | | | | | |
| **OUTCOMES:** | | | | | | | | | | | |
| **Presence of CPSP** | | | | | | | | | | X | X |
| **SF-MPQ-2** | | | | | | | | | | X | X |
| **BPI** | | | | | | | | | | X | X |
| **QoR-15** | X | | | | | | X | | X | | |
| **Intraoperative morphine equivalent** | | X | | | | | | | | | |
| **NRS score** | | X | X | X | X | X | X | X | X | | |
| **ASV time to spontaneous breathing** | | X | | | | | | | | | |
| **Time to first morphine rescue** | | X | X | X | X | X | X | X | X | | |
| **Postoperative morphine consumption** | | X | X | X | X | X | X | X | X | | |
| **Patient satisfaction** | | X | X | X | X | X | X | X | X | | |
| **Use of rescue antiemetics** | | X | X | X | X | X | X | X | X | | |
| **Nausea/vomiting** | | X | X | X | X | X | X | X | X | | |
| **ICU and hospital stay** | | | | | | | | | | X | |

**Fig 1. Assessments overview.** NRS, numerical rating scale; ASV, adaptive support ventilation; QoR-15, Quality of Recovery questionnaire; BPI, Brief Pain Inventory; SF-MPQ-2, Short-Form McGill Pain Questionnaire.

During the hospital stay, the following data will be retrieved from the hospital electronic record system: patient characteristics (age, gender, and EuroScore), the specific type of surgery performed, the duration of both the surgery and cardiopulmonary bypass, the time taken for Adaptive Support Ventilation (ASV) weaning to achieve spontaneous breathing, occurrences of nausea and vomiting along with any administered rescue antiemetics, and the total length of stay in both the ICU and the hospital (Fig 1).

## Sample size calculation

Sample size was calculated using G*Power software version 3.1.9.3 (Kiel University, Kiel, Germany), based on the QoR-15 score at 24 hours postoperatively – the primary outcome. The minimum clinically important difference (MCID) for the QoR-15 score is eight points [40], and the typical standard deviation (SD) ranges from 10 to 16 [41,42]. Assuming a two-sided type I error of 0.05, type II error of 0.2, and a population variance of 144 (SD = 12), a sample size of 36 per group is required. Allowing for a 20% dropout rate as a result of loss to follow up after patients discharge, a total of 96 patients (48 patients per group) will provide 80% power to detect a mean difference of ≥8 points in the QoR-15 score at 24 hours between the two groups.

## Data analysis

All outcomes will be analyzed and reported on an intention-to-treat basis, with patients analyzed according to their randomized group regardless of protocol adherence. A secondary per protocol analysis will be conducted for patients who do not adhere fully to the study protocol. Given the repeated measures of pain scores over time, which are correlated, Generalised Estimating Equation (GEE) models will be used to assess the time effects of postoperative analgesia. Chronic postsurgical pain outcomes will be analyzed using analysis of covariance (ANCOVA) to adjust for baseline differences and improve the precision of between-group comparisons. Baseline and 24-hour QoR-15 scores will be used as covariates for chronic postsurgical pain outcomes at three months and beyond to account for the potential influence of early recovery on chronic postsurgical pain perception. Categorical data will be presented using counts and their corresponding percentages. Continuous variables, will be reported as either their mean with standard deviation or their median with interquartile range, a choice that will depend on the normality of the data as determined by a Shapiro-Wilk's test. Between-group comparison will be conducted using the independent sample $t$-test for parametric data and Mann-Whitney U test for non-parametric data. Categorical variables will be compared using the Chi-square test. Data analyses will be performed using SPSS 27.0 (IBM Corp, Armonk, NY), and GEE modelling will be conducted using Stata V.14 (Statam College Station, Texas, USA), with a Gaussian distribution, identify-link function, exchangeable correlation structure, and robust standard errors. A P-value of <0.05 will be considered statistically significant, without adjusting for multiple comparisons.

## Ethics, data management and dissemination

The day before surgery, potential participants will be screened, and the study risks and benefits will be fully explained. Each participant must provide written informed consent, understanding their voluntary participation and the freedom to withdraw at any time without repercussions. A unique identifier code will be assigned to each participant for consistent use throughout the study. Research team members, specially trained in data entry, will input all information into an electronic system, with a second team member verifying for accuracy. The research team will conduct weekly monitoring of data collection and study conduct to ensure consistent adherence to protocols. All data will remain confidential, secured on password-protected computers and in locked cabinets within secured offices in the Department of Anesthesia and Intensive Care. Digital files will be permanently deleted and paper documents shredded five years post-study. Only aggregated group data will be published, with individual data accessible solely by study investigators. Ethical approval has been secured from the Joint University of Hong Kong-New Territories East Cluster Clinical Research Ethics Committee. The study will strictly comply with local laws, the Declaration of Helsinki, the International Council for Harmonization

of Technical Requirements for Pharmaceuticals for Human Use (ICH) Good Clinical Practice guidelines, and institutional policies. The safety of participants in this study will be monitored by an independent Trial Management Committee (TMC), comprising clinicians and external researchers. The research team is responsible for logging all adverse events related to the study drug and promptly reporting them to the TMC. The TMC will conduct a review of all reported events within 48 hours and address them during scheduled meetings. A predefined safety threshold is in place to unblind the study results to the TMC if a significant different in cumulative adverse events emerges between the two study groups (specifically, if the relative risk of adverse events in one group is greater than three and the interventional group has accumulated more than 20 adverse events). Should the interventional group demonstrate a higher adverse event rate, the trial will be immediately terminated. Examples of potential adverse events include hypotension requiring vasopressors, respiratory distress necessitating oxygen supplementation, pneumothorax, and local anesthetic toxicity. Study investigators will have the right to utilize the collected data. Furthermore, raw and summarized data will be made available upon reasonable request, with strict measures to maintain confidentiality and protect individual privacy. The study findings will be presented at international conferences and published in peer-reviewed journals.

## Discussion

Chronic postsurgical pain (CPSP) remains a significant and often debilitating complication following cardiac surgery, severely impacting patients' daily functioning, health-related quality of life, and contributing to increased healthcare costs [1–4,8,9]. Existing evidence highlights the crucial role of acute postoperative pain control in mitigating the development of CPSP. While various pain management strategies exist – including opioids, non-opioids, and multimodal analgesia – each has its limitations. Furthermore, current multimodal analgesia approaches, particularly for sternotomy pain, demonstrated limited efficacy [13]. Regional anesthesia techniques offer a promising avenue for enhanced pain control in cardiac surgery [14–17]. However, established neuraxial techniques such as epidural and intrathecal analgesia, despite their demonstrated efficacy, carry a potential risk of epidural haematoma in the context of systemic heparinization during cardiac surgery. Recent systematic reviews on paraspinal and chest wall blocks suggest potential benefits in reducing pain and opioid consumption, but are constrained by small sample sizes and methodological heterogeneity [16]. Techniques such as paravertebral blocks carry risks of pleural puncture and pneumothorax, while erector spinae plane block (ESPB) has shown inconsistent efficacy due to unpredictable local anesthetic spread [22,23]. Our own local experience with deep parasternal intercostal plane block also failed to demonstrate superior analgesia compared to conventional opioid-based approaches, despite a significant reduction in intraoperative opioid use [26]. This may be due to the limited duration of the block, which was insufficient to cover the prolonged surgical and postoperative recovery period. In addition, parasternal fascial plane blocks offer limited somatic (T2-T6) and visceral coverage.

This study addresses a critical research gap by investigating the efficacy of intertransverse process block, a novel regional technique, for pain management in cardiac surgical patients. ITPB offers several potential advantages over existing techniques. Magnetic resonance imaging studies have demonstrated consistent spread of local anesthetic to the intercostal and paravertebral spaces, neural foramina, and epidural space, suggesting the potential for comprehensive somatic and visceral analgesia [31]. Its anatomical location, relatively distant from major vessels and the neuraxial space, may reduce the risks of complications such as epidural haematoma and pleural puncture associated with other paraspinal and neuraxial regional techniques. Furthermore, the closer proximity of ITPB to the neural foramina and epidural space may offer more consistent and reliable analgesia compared to techniques like ESPB. With limited data available on the use of ITPB in cardiac surgery [43], we design this trial to establish its baseline analgesic efficacy. QoR-15 score at 24 hours is chosen as our primary outcome, rather than the long-term incidence of CPSP. The rationale for this decision is that QoR-15 is a validated tool that measures the quality of patient's early recovery, which is a strong predictor for long-term outcomes, including the development of CPSP. By first demonstrating that ITPB can improve early recovery and acute pain control, we can generate preliminary data vital for a larger, future trial powered to specifically study the long-term effects

of ITPB on CPSP. This approach ensures a foundational understanding of the technique's benefits before committing to a more resource-intensive, long-term study. Ultimately, this trial will be crucial for optimizing pain management strategies and enhancing recovery after cardiac surgery, addressing a critical unmet need in patient care.

## Supporting information

**S1 File. SPIRIT checklist.**
(DOCX)

**S2 File. Project submitted to the ethics committee.**
(PDF)

## Acknowledgments

We would like to thank all the cardiac anesthesiologists, surgeons, nurses who contributed and participated in this study, in particular, Ms Catherine Sze Yin WONG, Ms Sui King WONG, Ms Fung Yi LAI, and Ms Ying Ying CHIU.

## Author contributions

**Conceptualization:** Henry Man Kin Wong, Ranjith Kumar Sivakumar, Wai Tat Wong, Kwok Ming Ho.

**Data curation:** Henry Man Kin Wong.

**Formal analysis:** Henry Man Kin Wong.

**Methodology:** Henry Man Kin Wong, Ranjith Kumar Sivakumar, Kwok Ming Ho.

**Project administration:** Henry Man Kin Wong.

**Supervision:** Henry Man Kin Wong, Kwok Ming Ho.

**Writing – original draft:** Henry Man Kin Wong.

**Writing – review & editing:** Henry Man Kin Wong, Ranjith Kumar Sivakumar, Wai Tat Wong, Albert Kam Ming Chan, Zion Ho Sum Yeung, Pik Yu Chen, Sherry Tsz Wai Tang, Mandy Hiu Man Chu, Randolph Hung Leung Wong, Kwok Ming Ho.

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
