## [Decision Letter · Decision Letter 0]

29 Aug 2025

Dear Dr. WONG,

Thank you for submitting your manuscript to PLOS ONE. After careful consideration, we feel that it has merit but does not fully meet PLOS ONE’s publication criteria as it currently stands. Therefore, we invite you to submit a revised version of the manuscript that addresses the points raised during the review process.

**ACADEMIC EDITOR:**

We look forward to receiving your revised manuscript.

Kind regards,

Regina (Rianne) L.M. van Boekel

Academic Editor

PLOS ONE

Journal Requirements:

Reviewers' comments:

Reviewer's Responses to Questions

**Comments to the Author**

1. Does the manuscript provide a valid rationale for the proposed study, with clearly identified and justified research questions?

Reviewer #1: Yes

Reviewer #2: Yes

2. Is the protocol technically sound and planned in a manner that will lead to a meaningful outcome and allow testing the stated hypotheses?

Reviewer #1: Yes

Reviewer #2: Yes

3. Is the methodology feasible and described in sufficient detail to allow the work to be replicable?

Reviewer #1: Yes

Reviewer #2: Yes

4. Have the authors described where all data underlying the findings will be made available when the study is complete?

Reviewer #1: Yes

Reviewer #2: Yes

5. Is the manuscript presented in an intelligible fashion and written in standard English?

Reviewer #1: Yes

Reviewer #2: Yes

You may also provide optional suggestions and comments to authors that they might find helpful in planning their study.

Reviewer #1: Thank you for the opportunity to review this interesting and relevant study. I believe the topic has clinical value and the manuscript is overall well written. However, I have several minor suggestions that should be addressed before acceptance:

-In a study investigating the effect of a block on chronic pain, it is not entirely clear why Quality of Recovery (QoR) score was chosen as the primary outcome rather than the incidence of chronic pain itself. Please clarify the rationale and ensure alignment between the study aim and outcome selection.

-I recommend defining chronic postsurgical pain according to the ICD-11 classification.

Suggested reference:

Schug SA, Lavand'homme P, Barke A, et al. The IASP classification of chronic pain for ICD-11: chronic postsurgical or posttraumatic pain. Pain. 2019;160(1):45–52. doi:10.1097/j.pain.0000000000001413

-The authors should consider and discuss recent studies specifically addressing the impact of ESP block on chronic pain after cardiac surgery, which are highly relevant:

Dost B, Sarıkaya Ozel E, Kaya C, et al. Incidence of chronic postsurgical pain after cardiac surgery and the effect of bilateral erector spinae plane block: a randomized controlled trial. Reg Anesth Pain Med. Published online May 7, 2025. doi:10.1136/rapm-2025-106591

Also suggested references;

Yu X, Liu C. Intertransverse process block: A narrative review. J Clin Anesth. 2025;104:111857. doi:10.1016/j.jclinane.2025.111857

Dost B, Karapinar YE, Karakaya D, et al. Chronic postsurgical pain after cardiac surgery: A narrative review. Saudi J Anaesth. 2025;19(2):181–189. doi:10.4103/sja.sja_829_24

-The section currently titled “Ultrasound block placement” could be renamed to a more precise heading, for example “Ultrasound-guided block technique”

Reviewer #2: Interesting paper SOme issues should be addressed, although overall a very well written protocol

1) why authors choose to do this study single center?

2) it should be added who will perform the blocks

3) primary and secondary end points should be stated , also because sample size is computed on primary E.P.

4) blinding should be better detailed

**Do you want your identity to be public for this peer review?** For information about this choice, including consent withdrawal, please see our Privacy Policy

Reviewer #1: No

Reviewer #2: **Yes: ** Fabrizio D'Ascenzo

---

## [Author Response · Author response to Decision Letter 1]

8 Sep 2025

Reviewer 1

1. In a study investigating the effect of a block on chronic pain, it is not entirely clear why Quality of Recovery (QoR) score was chosen as the primary outcome rather than the incidence of chronic pain itself. Please clarify the rationale and ensure alignment between the study aim and outcome selection

Thanks for the insightful feedback. We appreciate reviewer’s concern regarding the selection of primary outcome measure. Both quality of recovery and chronic postsurgical pain are important patient-related outcomes in cardiac surgery. In the absence of data on analgesic efficacy of the intertransverse process block (ITPB) in cardiac surgery, it is a crucial first step in establishing baseline analgesic data for this novel technique.

We have selected Quality of Recovery-15 (QoR-15) score at 24 hours postoperatively as our primary outcome measure because it is a reliable, validated metric the captures multidimensional patient experience related to postsurgical recovery. The intensity and quality of early recovery are established predictors of long-term outcomes, including the development of chronic postsurgical pain [1]. Superior analgesia and patient comfort in the immediate postoperative period, as captured comprehensively by the QoR-15, are believed to mitigate the central sensitization and neuroplastic changes that underlie the transition from acute to chronic pain. By establishing the efficacy of ITPB in managing acute pain and improving early recovery, the study provides vital preliminary data that can inform the design and power calculations for subsequent larger trials specifically targeting chronic postsurgical pain as the primary outcome.

We have clarified this rationale in the revised manuscript (Lines 457-465) to better highlight the study’s priori role as a foundational investigation for ITPB in cardiac surgery. The title of the manuscript is also revised (Lines 4-6), for both the manuscript and the trial registration, to better align with the aim of the study.

Chronic postsurgical pain outcomes at three months and beyond will be analyzed both with and without adjusting for QoR at baseline and in the immediate postoperative period, to account for potential influence of early recovery on chronic postsurgical pain perception (Lines 365-369).

References:

[1] Maurice-Szamburski A, Bringuier S, Auquier P, Capdevila X. From pain level to pain experience: redefining acute pain assessment to enhance understanding of chronic postsurgical pain. Br J Anaesth. 2024 Nov;133(5):1021-1027. doi: 10.1016/j.bja.2024.08.003

2. I recommend defining chronic postsurgical pain according to the ICD-11 classification.

Suggested reference:

Schug SA, Lavand'homme P, Barke A, et al. The IASP classification of chronic pain for ICD-11: chronic postsurgical or posttraumatic pain. Pain. 2019;160(1):45–52. doi:10.1097/j.pain.0000000000001413

Thanks for the recommendation. The definition of chronic postsurgical pain in the manuscript text has been revised (Lines 292-294) with the suggested reference included.

3. The authors should consider and discuss recent studies specifically addressing the impact of ESP block on chronic pain after cardiac surgery, which are highly relevant:

Dost B, Sarıkaya Ozel E, Kaya C, et al. Incidence of chronic postsurgical pain after cardiac surgery and the effect of bilateral erector spinae plane block: a randomized controlled trial. Reg Anesth Pain Med. Published online May 7, 2025. doi:10.1136/rapm-2025-106591

Also suggested references;

Yu X, Liu C. Intertransverse process block: A narrative review. J Clin Anesth. 2025;104:111857. doi:10.1016/j.jclinane.2025.111857

Dost B, Karapinar YE, Karakaya D, et al. Chronic postsurgical pain after cardiac surgery: A narrative review. Saudi J Anaesth. 2025;19(2):181–189. doi:10.4103/sja.sja_829_24

Thanks for the suggestions. The recent study describing the effect of ESP block on chronic pain after cardiac surgery has been included in the manuscript (Lines 143-144) and appropriately cited. The review article on intertransverse process block is also appropriately cited in the manuscript.

4. The section currently titled “Ultrasound block placement” could be renamed to a more precise heading, for example “Ultrasound-guided block technique”

Thanks for the comment. The section title has been revised as suggested.

Reviewer 2

1. Why author choose to do this study single center?

Thank you for the question. A primary reason for doing this study single center is the cost and feasibility. This study is only supported by internal funding from my department. A single center design is therefore significantly less expensive and require fewer resources for administration and coordination, making it a practical choice. There are three cardiac surgical centers in Hong Kong, with slight differences in practice for both surgery and anaesthesia. Moreover, intertransverse process block is a relatively new regional technique that requires training and expertise. By confining the study to one location, we can ensure consistency in clinical practice and staff training. This helps reduce the risk of confounders that can arise from different institutional cultures or techniques, benefiting internal validity, where the observed effects are more likely due to the intervention itself rather than external factors. This is important for a preliminary study without much baseline data from the literature.

We acknowledge the limitation of a single-center design in generalizability. We hope to generate initial findings from this study which supports substantiation to a large-scale study in the future.

2. It should be added who will perform the blocks.

Thank you for the comment. We have indicated in the “Ultrasound-guided block technique” section that the study intervention will be performed by an experienced anesthesiologist who completed more than fifty successful ITPB procedures and takes no further part in the study.

3. Primary and secondary end points should be stated, also because sample size is computed on the primary E.P.

Thank you for the comment. The primary outcome is the Quality of Recovery-15 (QoR-15) score at 24 hours after tracheal extubation. The secondary outcomes include acute pain scores, time to first morphine rescue, patient satisfaction, postoperative morphine requirements, opioid-related side effects, and time to extubate. Chronic pain parameters such as incidence of chronic postsurgical pain, and the pain interferences will be recorded as secondary outcomes.

We have specifically highlighted and clearly stated our primary and secondary outcomes in the manuscript.

4. Blinding should be better detailed.

Thank you for the comments. The blinding details, including preparation of the randomization envelopes, the study syringes, and the personnels involved in the preparation are revised with better details (Lines 210-215)

---

## [Decision Letter · Decision Letter 1]

10 Sep 2025

Intertransverse process block to improve quality of recovery and pain management in cardiac surgery: protocol for a double-blinded randomized controlled trial

PONE-D-25-36620R1

Dear Dr. WONG,

We’re pleased to inform you that your manuscript has been judged scientifically suitable for publication and will be formally accepted for publication once it meets all outstanding technical requirements.

Kind regards,

Regina (Rianne) L.M. van Boekel

Academic Editor

PLOS ONE

Additional Editor Comments (optional):

Reviewer #1:

Reviewer #2:

Reviewers' comments:

Reviewer's Responses to Questions

**Comments to the Author**

1. Does the manuscript provide a valid rationale for the proposed study, with clearly identified and justified research questions?

Reviewer #1: Yes

Reviewer #2: Yes

2. Is the protocol technically sound and planned in a manner that will lead to a meaningful outcome and allow testing the stated hypotheses?

Reviewer #1: Yes

Reviewer #2: Yes

3. Is the methodology feasible and described in sufficient detail to allow the work to be replicable?

Reviewer #1: Yes

Reviewer #2: Yes

4. Have the authors described where all data underlying the findings will be made available when the study is complete?

Reviewer #1: Yes

Reviewer #2: Yes

5. Is the manuscript presented in an intelligible fashion and written in standard English?

Reviewer #1: Yes

Reviewer #2: Yes

You may also provide optional suggestions and comments to authors that they might find helpful in planning their study.

Reviewer #1: The authors have undertaken substantial revisions in response to the reviewers’ and editors’ comments. The manuscript has been considerably improved in terms of clarity, methodological detail, and overall presentation. I would like to congratulate the authors on their efforts and their commitment to enhancing the quality of the work.

Reviewer #2: All comments have been addressed and authors should be complimented for performing such a relevant study

**Do you want your identity to be public for this peer review?** For information about this choice, including consent withdrawal, please see our Privacy Policy

Reviewer #1: No

Reviewer #2: **Yes: ** Fabrizio d'ascenzo

---

## [Editor Report · Acceptance letter]

PONE-D-25-36620R1

PLOS ONE

Dear Dr. WONG,

I'm pleased to inform you that your manuscript has been deemed suitable for publication in PLOS ONE. Congratulations! Your manuscript is now being handed over to our production team.

Kind regards,

on behalf of

Dr. Regina (Rianne) L.M. van Boekel

Academic Editor

PLOS ONE